# Machine Learning-Enhanced Estimation of Cellular Protein Levels from Bright-Field Images

**DOI:** 10.3390/bioengineering11080774

**Published:** 2024-07-31

**Authors:** Takeshi Tohgasaki, Arisa Touyama, Shohei Kousai, Kaita Imai

**Affiliations:** 1FANCL Research Institute, FANCL Corporation, 12-13 Kamishinano, Totsuka-ku, Yokohama 244-0806, Japan; arisa1604@fancl.co.jp; 2Cytoronix Inc., 7-7 Shinkawasaki, Saiwai-ku, Kawasaki 212-0032, Japan; kousai@cytoronix.com (S.K.); imai@cytoronix.com (K.I.)

**Keywords:** machine learning, artificial intelligence, keratinocyte, live cell imaging, biomarker

## Abstract

In this study, we aimed to develop a novel method for non-invasively determining intracellular protein levels, which is essential for understanding cellular phenomena. This understanding hinges on insights into gene expression, cell morphology, dynamics, and intercellular interactions. Traditional cell analysis techniques, such as immunostaining, live imaging, next-generation sequencing, and single-cell analysis, despite rapid advancements, face challenges in comprehensively integrating gene and protein expression data with spatiotemporal information. Leveraging advances in machine learning for image analysis, we designed a new model to estimate cellular biomarker protein levels using a blend of phase-contrast and fluorescent immunostaining images of epidermal keratinocytes. By iterating this process across various proteins, our model can estimate multiple protein levels from a single phase-contrast image. Additionally, we developed a system for analyzing multiple protein expression levels alongside spatiotemporal data through live imaging and phase-contrast methods. Our study offers valuable tools for cell-based research and presents a new avenue for addressing molecular biological challenges.

## 1. Introduction

Analyzing the collective movement of cell populations alongside the behaviors of individual cells, including gene and protein expression, holds significant promise for advancing our understanding of various phenomena within biological systems. Cells demonstrate a spectrum of behaviors encompassing migration, apoptosis, division, differentiation, inflammatory responses, aging, and cancer development, while orchestrating internal gene expression, protein synthesis, and biochemical processes. Moreover, intercellular communication, whether direct or indirect, is pivotal in living organisms, facilitating diverse biological processes [1,2]. Minority cell populations have been documented as playing pivotal roles within cellular communities [3,4,5,6,7]. To elucidate the significance of these minority cells, imaging techniques with expansive field views and high spatial resolution have been developed [8]. With age, senescent cells accumulate in the body, secreting pro-inflammatory proteins known to induce the senescence-associated secretory phenotype, impacting surrounding tissues and fostering various age-related ailments, including cancer [9,10,11,12,13,14].

From this standpoint, comprehending cellular systems requires not only understanding the behavior of individual cells or the average behavior of populations, but also integrating population behavior with that of individual cells. However, simultaneous monitoring of individual cell behaviors within populations, alongside the expression of numerous genes and proteins over time, presents technical challenges. Although live-cell imaging enables observation of cellular dynamics, it is cumbersome to concurrently capture the expression of multiple genes and proteins. Conversely, single-cell RNA sequencing offers comprehensive insights into gene expression profiles across thousands to tens of thousands of cells, facilitated by various developed methodologies [15]. Nonetheless, these techniques are invasive and constrained in their ability to provide spatiotemporal information.

Advancements in image analysis technology utilizing artificial intelligence (AI) have enabled automated recognition of specific structures within images and estimation of various parameters from a single image [16,17]. Digital staining, for instance, permits the delineation of organelle regions, such as mitochondria and nuclei, within cells from bright-field images through deep learning algorithms [18]. Previously, we devised an AI model capable of identifying individual epidermal stratum corneum cell regions and estimating biomarker quantities [19]. Moreover, AI-driven cell analysis has seen significant advancements in recent years, encompassing identification of senescent cells, drug screening, and evaluation of responses to regenerative medicine [20,21,22].

Here, our objective was to pioneer a technology capable of estimating immunofluorescence images of intracellular proteins from bright-field images utilizing machine learning, thereby integrating bright-field and immunofluorescent-stained images of individual cells. The ability to predict protein immunofluorescence-staining images from bright-field images holds promise in estimating expression levels of diverse proteins from a single bright-field image by iteratively training for various proteins. Given the minimally invasive nature of acquiring bright-field images, this technology could facilitate simultaneous acquisition of intracellular protein expression and spatiotemporal information.

## 2. Materials and Methods

### 2.1. Cell Culture

Normal human epidermal keratinocytes (NHEKs) of Caucasian origin (Lonza, Basel, Switzerland) were cultured and utilized for machine learning purposes. To encompass various cell states, NHEKs were cultured with different types of media: CnT-PR (CELLnTEC, Bern, Switzerland), 0.001–0.5% sodium dodecyl sulfate (SDS) in CnT-PR, 0.2–100% keratinization-enhancing buffer CnT-PR-3D (CELLnTEC, Bern, Switzerland), or senescence-promoting medium CnT-AG2 (CELLnTEC) in CnT-PR. For high-throughput observation, NHEKs were seeded in a 96-well plate (Greiner Bio-One GmbH, Oberösterreich, Austria) and cultured with each medium for 48 h at 37 °C and 5% CO_2_.

### 2.2. Immunostaining

NHEKs were fixed with 4% paraformaldehyde in PBS for 1 h at 4 °C, washed with 0.05% Tween 20 in PBS, and then blocked with StartingBlock™ blocking buffer (Thermo Fisher Scientific, Waltham, MA, USA) at 37 °C. Following blocking, several biomarker proteins within the cells were labeled using the following antibodies diluted in StartingBlock™ blocking buffer at a 1:500 dilution for 1 h at 37 °C. After washing with 0.05% Tween 20 in PBS, the cells were incubated in secondary antibody solutions (Alexa Fluor 488 goat anti-mouse IgG, Alexa Fluor 488 goat anti-rabbit IgG) at dilutions of 1:1000 for 1 h at 37 °C. Additionally, for nuclear and cellular morphology labeling, 4′,6-diamidino-2-phenylindole (DAPI; Thermo Fisher Scientific) and rhodamine phalloidin (Thermo Fisher Scientific) were utilized at a 1:5000 dilution for 30 min at 37 °C.

The primary antibodies were anti-human IL-1 alpha/IL-1F1 antibody (R&D Systems, Minneapolis, MN, USA), anti-human IL-6 antibody (R&D Systems), anti-human NF-kβ p65 antibody (Bioss Inc., Woburn, MA, USA), anti-human Waf1/Cip1/CDKN1A p21 antibody (Santa Cruz Biotechnology, Inc., Dallas, TX, USA), anti-human p53 antibody (Santa Cruz), anti-human Beta Galactosidase antibody (Proteintech, Rosemont, IL, USA), anti-human Hsp27 antibody (R&D Systems), anti-human Galectin-7 antibody (R&D Systems), and anti-human Park7/DJ-1 antibody (R&D Systems). The secondary antibodies were Alexa Fluor 488 goat anti-mouse IgG, Alexa Fluor 488 donkey anti-goat IgG, and Alexa Fluor 488 goat anti-rabbit IgG (Thermo Fisher Scientific).

### 2.3. Imaging for AI Model

Following staining, fluorescent and phase-contrast images (960 × 720 pixels) of the NHEKs in each well were captured using a BZ-810 fluorescence microscope (Keyence Corporation, Tokyo, Japan) equipped with a 20× dry lens. NHEKs cultured under the aforementioned conditions were observed in 180 frames from 60 wells for each biomarker. To ensure consistency, phase-contrast images of NHEKs before and after fixation in the same field of view were obtained. After capturing the cells, fixation was performed on the microscope stage, followed by the recapturing of the fixed cells.

### 2.4. Image Processing and Machine Learning

Each cell region in the captured phase contrast was extracted by binarizing the rhodamine phalloidin fluorescence images. The fluorescence intensity values representing each biomarker protein in the cells were normalized by the area of each extracted region and converted into an average value per pixel. To develop a machine learning model capable of estimating fluorescence intensity levels in each cell from phase-contrast images, the processed phase-contrast and fluorescence images were paired for learning (Figure 1a). The CNN/U-Net architecture was used for machine learning [23]. Figure 1b shows the convolutional network used in this study. The input picture was cropped to 256 by 256 pixels and then applied to the network. The loss of the output picture was calculated using the mean squared error function. To validate the accuracy of the constructed model in estimating biomarker protein levels, the model was tested on images not used during training, and a comparative evaluation was conducted between the estimated and actually stained images.

### 2.5. Time-Lapse Imaging

NHEKs were cultured in CnT-PR culture medium in a 96-well plate (Greiner Bio-One GmbH) and stained with Hoechst 33342 (Thermo Fisher Scientific) and CellMask™ (Thermo Fisher Scientific) to label the nucleus and cell membrane. After staining, the NHEKs were subjected to live imaging using a BZ-810 fluorescence microscope equipped with a culture device STXG-KIWXA20I-SET (Tokai Hit Co., Ltd., Shizuoka, Japan). Fluorescence and phase-contrast images of NHEKs were captured every 30 min for 109.5 h using a 20× dry lens. Subsequently, the AI model developed in Section 2.4 was applied to the obtained time-lapse images to estimate the expression levels of nine proteins in each NHEK.

### 2.6. Statistical Analysis

Correlation analysis was conducted using Pearson’s product-moment correlation test. All statistical analyses were performed using Bell Curve for Excel ver. 7.0 (Social Survey Research Information Co. Ltd., Tokyo, Japan). Results were considered statistically significant at *p* < 0.05.

## 3. Results

### 3.1. Evaluation of Protein Estimation Accuracy Using Machine Learning

To assess the accuracy of intracellular biomarker protein expression-level estimation from our AI model, we compared actual stained images with images estimated by the machine learning model for the nine biomarkers studied (interleukin 1 alpha [IL-1α], interleukin-6 [IL-6], nuclear factor-kappa beta [NF-κB], p53, p21, galactosidase beta 1 [GLB1], heat shock protein 27 [HSP27] galectin-7 [GAL7], and PARK7/DJ-1). Visual qualitative evaluation revealed a consistent relationship between the intensity levels of each biomarker protein in normal human epidermal keratinocytes (NHEKs), albeit with some observed differences (Figure 2). Additionally, we conducted correlation analysis between the average intracellular fluorescence brightness values measured from stained images and those estimated by AI. Significant correlations were observed for all proteins (IL-1α: *r* = 0.836, *p* < 0.001; IL-6: *r* = 0.726, *p* < 0.001; NF-κB: *r* = 0.622, *p* < 0.001; p53: *r* = 0.440, *p* < 0.001; p21: *r* = 0.530, *p* < 0.001; GLB1: *r* = 0.801, *p* < 0.001; HSP27: *r* = 0.743, *p* < 0.001; DJ-1: *r* = 0.845, *p* < 0.001; GAL7: *r* = 0.810, *p* < 0.001; Figure 3). Notably, the estimated protein intensities were higher than the measured intensities. The scatterplots for IL-1α and IL-6 show a steep increase from a flat tail for both cytokines. Their measured values (approximately 20 and 40) were at the lower detection limit of staining. Values lower than this can thus not be analyzed using staining. However, the AI expressed differences in the signal values between cells based on the learned algorithm that were also below the detection limit for actual staining.

We confirmed that the AI model detected differences in expression intensity even in cells with low measured intensity, particularly for IL-1α and IL-6. Proteins primarily localized in the nucleus, such as p53, p21, and NF-κB, exhibited lower correlation coefficients than other markers.

### 3.2. Applicability of the Protein Expression Estimation Model to Live NHEKs before Fixation

We validated the AI model developed using the aforementioned method, which estimated the expression levels of nine proteins in fixed NHEKs, for application to living (unfixed) cells. To assess the disparities in the inference values of the AI model for NHEKs before and after fixation, we acquired phase-contrast and fluorescence images of the nucleus and cell membrane of NHEKs before and after fixation within the same field of view. The fluorescence values of the cell membrane exhibited no significant changes after fixation, and no alteration was observed in the cell recognition area (Figure 4a). Additionally, high correlations were observed between the inference values of each protein in NHEKs before and after fixation predicted by the AI model, except for GLB1 (IL-1α: *r* = 0.562, *p* < 0.001; IL-6: *r* = 0.891, *p* < 0.001; NF-κB: *r* = 0.678, *p* < 0.001; p53: *r* = 0.809, *p* < 0.001; p21: *r* = 0.841, *p* < 0.001; HSP27: *r* = 0.682, *p* < 0.001; /DJ-1: *r* = 0.766, *p* < 0.001; GAL7: *r* = 0.857, *p* < 0.001; Figure 4b). The absence of a correlation between the inferred values before and after the fixation of NHEKs for GLB1 indicated distinct inferred values (*r* = 0.149, *p* = 0.077). These results indicate that the AI model developed based on fixed cells can be applied to infer the expression levels of eight of the analyzed proteins in NHEKs before fixation, but not for GLB1.

### 3.3. Integration of Machine Learning Models into Live Imaging

The AI model was used to analyze time-lapse images of NHEKs, predicting the spatiotemporal behavior of cells and the expression levels of nine proteins (Appendix A). Time-lapse images revealed characteristic fluctuations in protein expression levels within NHEKs, including those influenced by cell movement or environmental conditions, those exhibiting random variations, and those showing negligible alterations. For instance, the expression levels of HSP27 and GAL7 increased as cell density increased, whereas the expression level of p21 was higher under sparse conditions than under confluent conditions. Notably, p53 expression was unrelated to cell density (Figure 5a). Conversely, p53 expression increased in response to cellular damage. Figure 5b illustrates time-lapse images before and after cell rupture, depicting a transient increase in p53 expression following damage.

## 4. Discussion

In this study, we developed a machine learning model to estimate immunofluorescence-staining images of intracellular proteins from phase-contrast images and validated its accuracy. We chose keratinocytes as our target owing to their flat, circular, and adherent nature, facilitating automatic recognition of individual cell regions. Additionally, keratinocytes exhibit a spectrum of behaviors including inflammation, aging, and differentiation [24,25,26]. For the verification of deep learning, we selected target proteins associated with differentiation (HSP27, GAL7), aging (p21, p53, GLB1), inflammation (IL-1a, IL-6, NF-κB), and antioxidant effects (DJ-1). In addition, these selected proteins are localized in the cytoplasm (HSP27, GLB1, GAL7, DJ-1) or nucleus (p21, p53), or they are secreted (IL-1a, IL-6, NF-κB), each with different characteristics.

Our results demonstrated significant correlations between the fluorescence values of actual staining and estimated images for all proteins. This suggests the feasibility of approximating fluorescence images from phase-contrast images. Notably, our machine learning model quantified low expression levels and fluorescence intensity in cells expressing IL-1a and IL-6, even below detection limits. That is, AI has the potential to estimate expression levels of proteins, even in a range below the detection limit, that cannot be analyzed using actual staining. However, we could not verify the accuracy of the estimated value due to the absence of actual staining values. Therefore, accurately converting these quantified differences at low intensities into values requires validation using highly sensitive quantitative methods. Moreover, proteins such as p21, p53, and NF-κB, specifically expressed in the nucleus, exhibited lower correlation coefficients than other proteins, indicating lower estimation accuracy. As our approach involved averaging fluorescence intensity across cells using image processing, locally expressed proteins within cells posed challenges in estimating actual local intensity, potentially leading to decreased accuracy. Refinement of the method to estimate intensity solely within the nucleus, rather than averaging across the cell body, from phase-contrast images may enhance estimation accuracy, especially for proteins localized in the nucleus. Furthermore, to assess the applicability of our constructed AI model to living cells prior to fixation, we estimated protein expression levels from images obtained before and after fixation. High correlations were observed between estimated values before and after fixation for all proteins except GLB1. This suggests the suitability of our method for cells before fixation. However, significant differences in estimated values before and after fixation for GLB1 indicate that the same model may not be applicable to living cells. Further investigation is warranted to elucidate this discrepancy, suggesting the need for machine learning using images stained with fluorescent probes for GLB1 before fixation.

We applied our AI model to time-lapse images of NHEKs to obtain time-series images of estimated expression levels for nine types of proteins. Analysis revealed relationships between cell spatiotemporal behavior and protein expression. For instance, HSP27 and GAL7 expression levels increased as cells became confluent, whereas p21 expression was higher in sparse states, and p53 expression was unrelated to cell density. Additionally, a transient increase in p53 expression was observed immediately after cell injury. These examples underscore the utility of our AI technology in analyzing relationships between protein expression and keratinocyte behavior in the surrounding environment.

Technologies for phase-contrast image-based intracellular protein expression-level estimation could be further enhanced. Although our attempt at estimation using phase-contrast images provides versatility and orientation-free imaging, other illumination methods, such as differential interference contrast and apodized phase contrast, also enable non-invasive imaging, potentially improving estimation accuracy by combining multiple images with diverse information. A technology leveraging deep learning for automatic estimation of organelle regions, such as mitochondria and nuclei, from phase-contrast images has been reported [18]. Although our study verified the capability of this technology to estimate average protein fluorescence levels in cells, the localization of each protein remains unexplored. This limitation stems from the assumption that protein molecules, being much smaller than organelles, cannot be visually annotated from phase-contrast images, thus hindering estimation of protein localization. However, estimating the localization of proteins specifically expressed in organelles, such as nuclear proteins, may be feasible.

## 5. Conclusions

Our study demonstrates that by iteratively using deep learning for various proteins, the expression-level estimation of numerous proteins is feasible, relying solely on a single bright-field image. Given the minimally invasive nature of obtaining bright-field images, long-term time-lapse imaging could be conducted, with our AI model applied to analyze spatiotemporal information alongside protein expression levels in individual cells. In addition, bright-field image-based simultaneous estimation of organelle localization and morphology enables cellular state monitoring. Our technique is effective for experimental systems using cultured cells and contributes to deepening our understanding related to the molecular cell biology underlying various life phenomena such as differentiation, cancer, and aging.

## Figures and Tables

**Figure 1 bioengineering-11-00774-f001:**
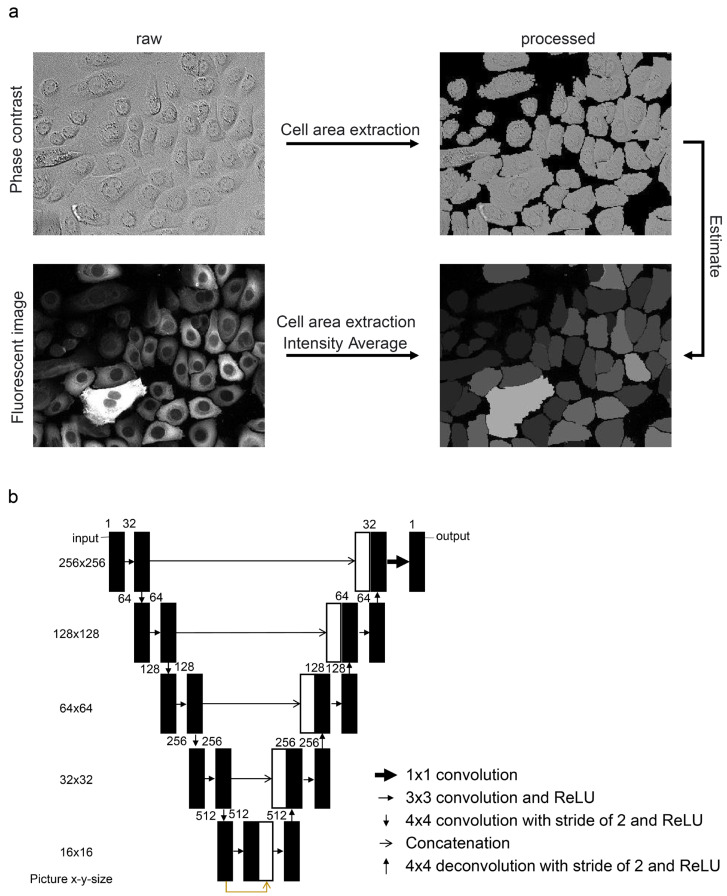
Image processing methods and machine learning schemes. (**a**) A phase-contrast image of NHEKs (upper left) alongside a processed image extracting cell areas by eliminating the background (upper right). The fluorescence immunostaining image on the bottom left displays Hsp27 as an example, accompanied by a processed image averaging the intensity within each cell. Learning was conducted to estimate the average fluorescence image at the bottom right of the phase-contrast image at the top right. (**b**) U-net architecture. Each box corresponds to a multi-channel map. The number of channels is denoted at the left top of the box. The picture size is provided in the left column. The arrows denote different operations. After each 3 × 3, 4 × 4, and 4 × 4 deconvolution, the rectified linear unit (ReLU) function was applied.

**Figure 2 bioengineering-11-00774-f002:**
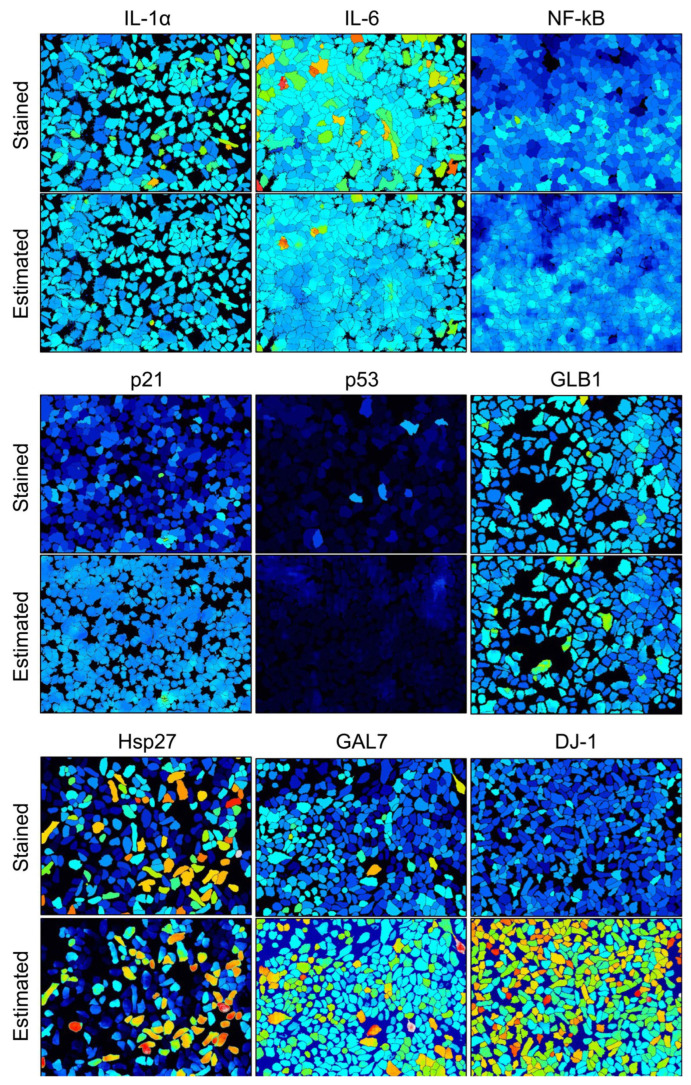
Actual stained image and estimated image. Fluorescent immunostaining images depicting the average intensity of intercellular proteins for each cell (**upper row**) and images estimated by the machine learning model (**lower row**). Brightness is represented on a thermoscale.

**Figure 3 bioengineering-11-00774-f003:**
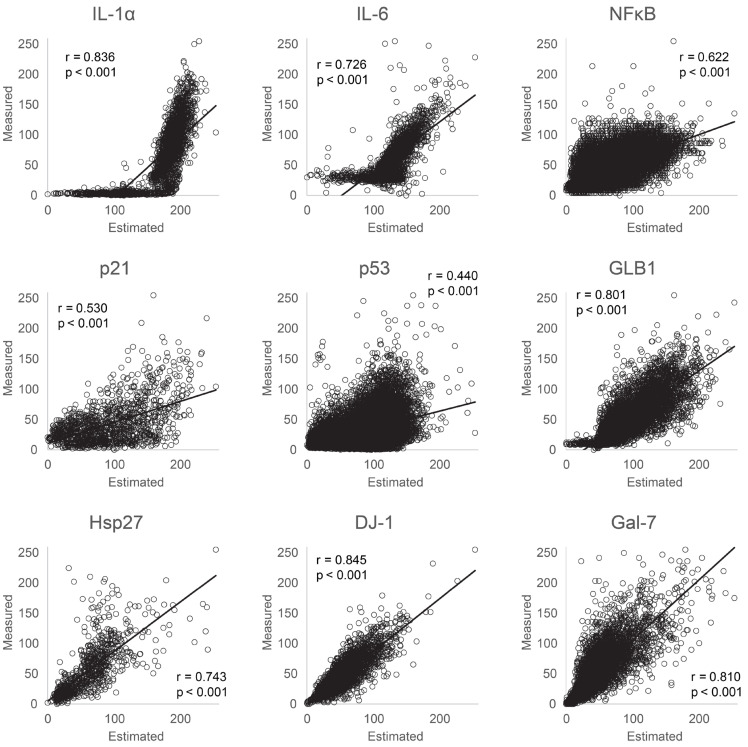
Correlation graph of protein expression levels in NHEKs between actual stained and estimated images. Correlation graphs illustrating the measured fluorescence levels against the estimated levels of each cell from the machine learning model are shown.

**Figure 4 bioengineering-11-00774-f004:**
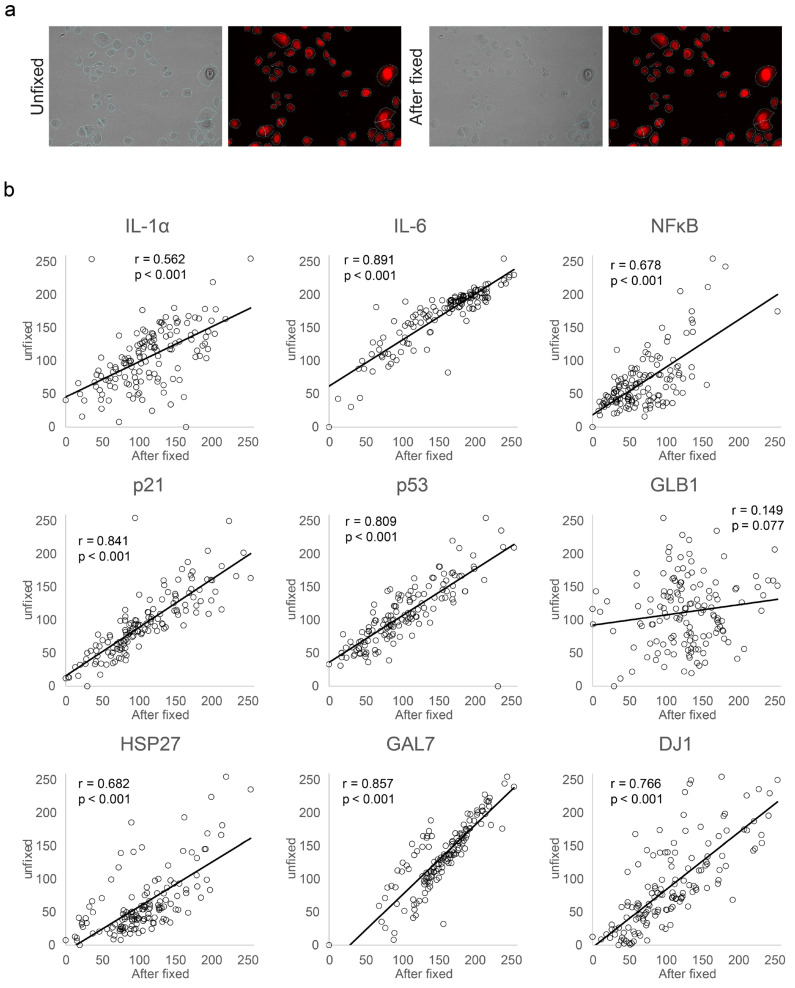
NHEK morphology and correlation graph of protein expression level before and after fixation treatment. (**a**) Phase-contrast and fluorescence images before and after fixation treatment. Phase-contrast image (left row) and cell membrane fluorescence image (right row). Recognized cell area outlined in blue. (**b**) Correlation graphs displaying intensity levels of each cell estimated by machine learning for cell images before and after fixation.

**Figure 5 bioengineering-11-00774-f005:**
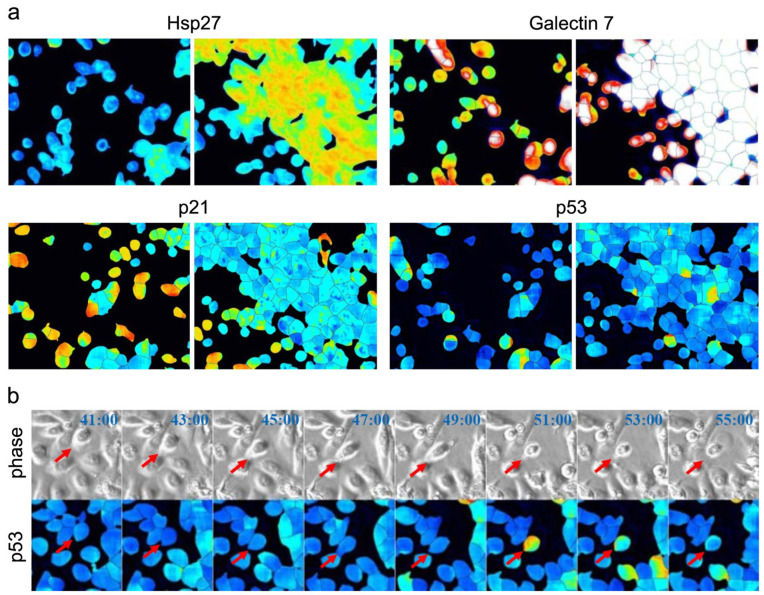
Protein expression levels in NHEKs exhibit characteristics for each protein. (**a**) Depiction of the relationship between cell density and the expression level of each protein. Images depict sparse cells (left) and dense cells (right). The images represent enlarged segments of Appendix A. (**b**) Time-lapse images of phase contrast and p53 estimated by AI. Cell behavior and shape were observed in phase-contrast images (upper row), accompanied by the expression level of p53 in the same field of view. The images represent enlarged segments of Appendix A, with time (blue characters) indicating the same time as Appendix A. Red arrows indicate before and after cell rupture, depicting a transient increase in p53 expression following damage.

## Data Availability

The original contributions presented in the study are included in the article/Appendix A, further inquiries can be directed to the corresponding author.

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
