# Peer review of "Machine Learning-Enhanced Estimation of Cellular Protein Levels from Bright-Field Images"

_bioengineering, 2024, doi:10.3390/bioengineering11080774_

Round 1
Reviewer 1 Report
Comments and Suggestions for Authors
The authors presented a machine learning algorithm to estimate cellular protein levels from bright-field images. They found that the AI model achieve favorable estimation of protein expression from solely a single bright-field image. The results and analysis appear intriguing and promising; however, there are several major issues need to be addressed before further consideration of this manuscript.
1. The codes and algorithms of this AI method were not shown in current draft. It is commonly required to share and distribute such information on Github or other open-access websites. Please provide this information.
2. In the correlation plots of Figure3 and Figure4, the authors should add the correlation values and p-value into each plot. Thus it can be clearer and more comparable between different proteins.
3. The English writing of this manuscript should be improved, as there are numerous sentences that are unclear and inaccurate. For example, in Abstract, the authors wrote “This breakthrough offers valuable tools for cell-based research…”, I would suggest revise ‘breakthrough’ to ‘Our study’. The authors may consider seeking assistance from editing service or a native English speaker to improve the language.
4. In Figure3 protein IL-1α, why the scatter plot shows a flat tail between 0 to 150, and then a steep increase from 200? Does this indicate the false positive estimation of the AI model? How would the authors address and improve this issue?
Comments on the Quality of English LanguageExtensive editing of English language required
Reviewer 2 Report
Comments and Suggestions for Authors
In this manuscript, authors have aimed to develop a novel method for determining a non-invasive, intracellular protein levels that are essential for understanding cellular phenomena. The results from this work has revealed the insights into gene expression, cell morphology, dynamics, and intercellular interactions.
The works is scientifically sound and manuscript is well-drafted.
This can be accepted in the present form.
Author Response
We thank the reviewer for commending our work.
Round 2
Reviewer 1 Report
Comments and Suggestions for Authors
The authors have addressed my comments and I endorse this paper for publication.